# Complex Multimorbidity and Incidence of Long-Term Care Needs in Japan: A Prospective Cohort Study

**DOI:** 10.3390/ijerph181910523

**Published:** 2021-10-07

**Authors:** Daisuke Kato, Ichiro Kawachi, Junko Saito, Naoki Kondo

**Affiliations:** 1Department of Family Medicine, Mie University Graduate School of Medicine, Tsu 514-8507, Japan; 2Department of Social and Behavioral Sciences, Harvard School of Public Health, Cambridge, MA 02115, USA; ikawachi@hsph.harvard.edu; 3Center for Public Health Science, National Cancer Center Japan, Tokyo 104-0045, Japan; j.junkosaito@gmail.com; 4Science Frontier Laboratory, Department of Social Epidemiology, Graduate School of Medicine and Public Health, Kyoto University, Floor 2, Yoshida-konoe-cho, Sakyo-ku, Kyotoshi, Kyoto 606-8315, Japan; kondo.naoki.0s@kyoto-u.ac.jp

**Keywords:** multimorbidity, complex multimorbidity, long-term care needs

## Abstract

Complex multimorbidity (CMM) has been proposed as a more nuanced concept of multimorbidity (MM). We sought to quantify the association of CMM and MM on the incidence of long-term care (LTC) needs in a cohort of older Japanese people. Our follow-up was based on a nationwide longitudinal cohort study of people aged over 65 years who were functionally dependent at baseline. Our outcome was incident LTC needs, based on certification under the Japanese LTC insurance scheme. We used both propensity score matching and inverse probability of treatment weights (IPTW) to compare individuals with and without MM versus CMM. A total of 38,889 older adults were included: 20,233 (52.0%) and 7565 (19.5%) adults with MM and CMM, respectively. In propensity-matched analyses, both MM (*n* = 15,666 pairs) and CMM (*n* = 7524 pairs) were statistically significantly associated with the six-year LTC insurance certification rate (MM, hazard ratio (HR) 1.07, 95% confidence interval (95%CI) 1.02–1.12; CMM, HR 1.10, 95%CI 1.04–1.16). Both MM and CMM were associated with a modest but statistically significantly higher rate of LTC insurance certification. These findings support the inclusion of multimorbidity in the assessment of LTC insurance needs, although the Japanese government currently has not adopted this.

## 1. Introduction

Globally, older people are suffering from multiple chronic diseases as they get older [1]. This multimorbidity (MM) is most commonly operationalized by summing the number of diseases that co-exist in the patient at the same time [2]. The co-presence of plural conditions interactively increases the risk of long-term outcomes such as the incidence of long-term care needs, mortality, reduced quality of life, and so on [2,3].

Attempts have been made to improve the measurement of multimorbidity, such as weighting each disease according to severity, as opposed to using a simple count [4]. A more recent advance is represented by the concept of “complex multimorbidity” (CMM) [5]. Instead of focusing on disease, CMM focuses on its superordinate concept, the body system. CMM is defined as a disorder of more than two different body systems simultaneously. CMM is theorized to demonstrate—to a greater extent than MM—the significance of comorbidity from a biomedical point of view, as diseases spanning multiple body systems would have a stronger impact on patient outcomes. Furthermore, from the methodological perspective, focusing on body system disorders may be a more reliable approach because while patients can misclassify their own individual conditions that overlap with respect to location and symptoms (e.g., osteoarthritis versus rheumatoid arthritis), CMM would be less affected by such misclassifications. In addition, CMM is also apt to be more reliable because it would minimize the impact of multiple diagnoses for a single condition provided by different clinicians. However, studies that directly compared the performance of MM versus CMM on patient outcomes have remained sparse. Furthermore, most such studies evaluating both MM and CMM were mainly based on descriptive statistics, not inferential statistics [6,7].

The national long-term care (LTC) insurance system in Japan was introduced in 2000 and all persons over 65 years in Japan are eligible for LTC as long as they are certified under the official insurance system [8]. About 20% of older adults in Japan use LTC services as of January 2020, and the number is on the rise [9]. The conditions associated with the greatest need for LTC, in descending order, are dementia, cerebrovascular disorders, frailty, fractures/falls, and joint disorders, all of which rise with age [10]. While it is possible for older patients with multimorbidity to achieve healthy aging via the appropriate provision of medical resources [11], the association between MM and LTC need is not well documented.

The LTC certification process consists of a two-step assessment procedure. Initially, trained local government staff visit the individuals in their homes in order to evaluate their need for LTC services based on questionnaires inquiring about their current physical and cognitive status. In the second step, the results of the home visit assessment are reviewed by a municipal panel of physicians, nurses, and social care workers to certify each applicant with respect to their eligibility for long-term care services.

Currently, the certification of LTC needs in Japan is determined only by the degree of physical and cognitive impairment, while the presence of MM is not considered. Conventional disease classifications that do not take into account MM may not adequately assess the need for LTC services and their provision because they only independently assess the impact of each disease [11].

Therefore, we prospectively examined the association of multimorbidity with the incidence of LTC needs for older adults in Japan with two types of multimorbidities (conventional MM and CMM).

## 2. Materials and Methods

### 2.1. Study Cohort

We used longitudinal data from the Japan Gerontological Evaluation Study (JAGES), a nation-wide cohort of adults aged 65 years or older established in 2010, which aims to establish a society of healthy longevity from the viewpoint of preventive medicine.

At baseline, self-administered questionnaires were mailed between August 2010 and January 2012, which covered 95,827 older adults sampled from 13 municipalities in 7 of the 47 prefectures in Japan. The municipalities were selected among the major islands of Japan (Hokkaido, Honshu, Kyushu, and Japan) except for Shikoku. In 9 municipalities, a census of the entire population aged over 65 years was conducted, while in the remaining 4 municipalities simple random sampling was performed based on the official residential registers maintained by municipal authorities. The Nihon Fukushi University Ethics Committee (no. 10-5), the National Center for Geriatrics and Gerontology (no. 992-2), and the Chiba University Ethics Committee (no. 2493) approved the parent JAGES protocol. Further details of the JAGES cohort have been provided elsewhere [12].

From the target population, 62,426 individuals participated in the survey (response rate, 65.1%). Of these, 56,687 individuals had valid information on ID number, sex, and age (valid response rate, 59.2%). Of the 56,687 valid respondents, 54,537 (96.2%) were successfully linked to the LTC insurance certification registers.

We excluded individuals whose functional disability status at baseline was unknown, or who were already receiving nursing care or home care assistance, or whose data about the history of present illness were missing. Finally, the number of participants in this analysis was 38,889. The study cohort flow diagram is shown in Figure 1.

### 2.2. Assessment of MM and CMM

The explanatory variables in this study were MM and CMM. First, MM was defined as the coexistence of two or more diseases. At baseline, JAGES surveyed 19 diseases. Among these, two symptoms on the JAGES survey, “Difficulty swallowing” and “Difficulty with bowel movement”, were excluded because they were strictly dysfunctions, not diseases. Therefore, this study included 17 diseases: “Heart disease (including arrhythmia)”, “Stroke”, “High blood pressure”, “Diabetes (including mild type)”, “Obesity”, “Dyslipidemia”, “Impaired vision”, “Gastrointestinal disease”, “Liver disease”, “Impaired hearing”, “Mental disease”, “Sleep problem”, “Osteoporosis”, “Joint disease/Neuralgia”, “Injury/Fracture”, “Cancer”, and “Respiratory disease”.

Second, CMM was defined as the simultaneous presence of body system impairment across more than two categories at baseline, e.g., circulatory (heart disease), endocrine–metabolic (diabetes), and gastrointestinal (liver disease). Surveyed diseases were classified into categories according to the affected body system [5].

For additional diseases surveyed by JAGES, new categories were created based on a previous study as needed [13]. Specifically, “Hearing loss” and “Gastrointestinal disorder” were newly added as categories. Furthermore, diseases regarding the “Nerve disorders” category were not collected in JAGES (See Table 1).

### 2.3. Endpoint Assessment

The endpoint used in this study was a six-year incidence of LTC insurance certification. The certification information was obtained by record linkage to the insurance registers from participating municipalities. Event time was measured in days until the LTC insurance certification.

### 2.4. Propensity Score Matching

The association between multimorbidity and downstream health outcomes (such as LTC needs) is likely to be confounded by a number of factors that are common prior causes of the exposure and outcome. Potential confounders include a wide array of socio-demographic variables, socioeconomic status, and health behaviors. Consequently, a comparison of outcomes among individuals with/without multimorbidity will be biased unless confounding is taken into account. Conventional multivariable regression may not adequately control for confounding if the exposure groups are not balanced with respect to the distribution of covariates.

Propensity score matching attempts to improve control for confounding by avoiding off-support inferences, i.e., comparison of treatment and control groups that are not counterfactually exchangeable [14].

To minimize the impact of confounding bias, we estimated the propensity scores for having MM/CMM by two-level logistic regression. In the calculation of the propensity score, 44 variables were included as potential confounders of treatment (in the instance, having MM/CMM): age, sex, smoking history, alcohol use, marital status, pension, dental health, working status, consumption of meat or fish/fruits or vegetables, education, city code, and so on (See Appendix A).

We assumed missing data as missing at random and imputed missing data using a bootstrapping expectation maximization with a bootstrapping algorithm [15]. Considering the small amount of missing data in the cohort data (around 5%), we analyzed 20 multiply imputed datasets [16] and combined all estimators by Rubin’s rule [17].

### 2.5. Matching

Participants with and without MM/CMM were matched using a nearest neighbor matching strategy with a 1:1 ratio [14]. The caliper width was equal to 0.2 times the standard deviation of the logit of the propensity scores [18].

### 2.6. Inverse Probability of Treatment Weighting (IPTW)

We additionally applied inverse probability of treatment weights to analyze the effect on MM/CMM for the LTC insurance certification [19]. The pseudo population generated by IPTW stabilizes the number of participants in the with/without MM group, and is useful for estimating the treatment effect with a lower mean squared error [20].

We adopted an average treatment effect (ATE) to examine the causal effects of MM/CMM and LTC insurance certification. In addition, we adjusted the ATE weight by multiplying the ATE weight by the proportion of each group as a whole (stabilized ATE weight) to deal with the excessively high IPTW when the PS was close to 0 (for the without MM/CMM group) or 1 (for the with MM/CMM group). This stabilized IPTW preserves the sample size of the original data and leads to appropriate control of the type I error rate [21].

### 2.7. Statistical Analysis

The C-statistic was quantified to evaluate the discriminatory ability of the propensity score model [22]. After the matching or IPTW, we compared the demographic data between the population with and without MM/CMM groups using the standardized difference. An absolute standardized difference > 0.1 was considered to indicate a significant imbalance of a baseline covariate [18].

Using a regression model, we compared the LTC insurance certification between the population with and without MM/CMM groups in both the propensity-score-matched cohort and the propensity-score-weighted cohort. Specifically, the cumulative incidence of LTC insurance certification was estimated using Kaplan–Meier curves. The difference in certification risk was analyzed by an event-weighted log-rank test [23]. The proportional hazards assumptions were examined (and found to be supported) by plotting the log–log survival curves. We estimated the hazard ratio for the certification risk of MM/CMM with regression models. MM/CMM was the only variable in the models because the cohort was balanced. Population-level hazard–ratio effects by propensity-score methods are more similar to the effects estimated in a randomized controlled trial than those by multivariable Cox regression [18].

All statistical analyses were performed with R software packages (version 4.0.1). All *p* values are two-tailed, and statistical significance was set at *p* = 0.05.

### 2.8. Ethical Considerations

The JAGES participants were informed that participation was voluntary and their consent to participate in the study was shown by the questionnaire’s return via mail. The Nihon Fukushi University Ethics Committee (no. 10-5), the National Center for Geriatrics and Gerontology (no. 992-2), and the Chiba University Ethics Committee (no. 2493) approved the parent JAGES protocol.

## 3. Results

### 3.1. Baseline Population Characteristics

After applying the selection criteria, 38,889 individuals were identified, including 20,233 (52.0%) individuals with MM and 7565 (19.5%) individuals with CMM. Appendix A summarize selected demographic characteristics of the study population with and without MM/CMM. Participants without MM were more likely to be younger, were more likely to have more teeth, and were more likely to spend more time walking. On the other hand, participants without CMM were more likely to be female and younger, were more likely to have more teeth, higher education, and meals with others, were more likely to be married and spend more time walking and engaging in social participation, and were more likely to have trust in and support from neighbors. The propensity score-matched/-weighted cohorts were well balanced, with all the standardized differences <10% (see Table 2 and Table 3).

### 3.2. MM Outcome

A total of 5588 (27.6%) LTC insurance certifications occurred in the population with the MM group and 3580 (19.2%) in the population without the MM group. The Kaplan–Meier plot of the cumulative incidence of LTC insurance certification is shown in Figure 2. The participants with MM had a shorter time to the LTC insurance certification than those without MM, which corresponds to a 7% higher risk of LTC insurance certification (hazard ratio, 1.07; 95%CI, 1.02 to 1.12; *p* = 0.006 by the log-rank test). The hazard ratios by IPTW were comparable in direction and significance to those from propensity score matching.

### 3.3. CMM Outcome

A total of 2508 (33.2%) LTC insurance certifications occurred in the population with the CMM group and 6660 (21.3%) in the population without the CMM group. The Kaplan–Meier plot of the cumulative incidence of LTC insurance certification is shown in Figure 3. The participants with CMM had a shorter time to LTC insurance certification than those without CMM, which corresponds to a 10% higher risk of LTC insurance certification (hazard ratio, 1.10; 95%CI, 1.04 to 1.16; *p* = 0.001 by the log-rank test). The hazard ratios using IPTW were comparable in direction and significance to those from propensity score matching.

Although the number of diseases and body system disorders that define MM and CMM has already been validated in previous studies [5], we also applied a more sensitive approach. That is, we analyzed the association between the number of diseases or body system disorders and LTC insurance certification by using linear regression analysis. However, the sensitivity analysis above did not affect our conclusions.

## 4. Discussion

To our knowledge, this is the first study to demonstrate that CMM is associated with the incidence of LTC needs. Our findings suggest that CMM was associated with a modestly elevated risk of LTC insurance certification among people aged over 65 years in Japan. This association was independent of other variables influencing LTC needs, including age, sex, years of education, alcohol use, smoking history, or socioeconomic status, which are already known to influence LTC needs and multimorbidity in older adults [24,25].

The results of our study suggest the importance of focusing on morbidity across several body systems (i.e., complex multimorbidity) in order to prevent disability. Furthermore, when combined with our previous findings [26], our results suggest that older individuals with MM/CMM are at increased risk of long-term care needs as well as higher mortality.

The HR for LTC certification was comparable for MM and CMM (i.e., their 95% confidence intervals largely overlapped). This finding is notable because few studies so far have compared the impact of MM versus CMM on patient outcomes. On the other hand, there are some emerging empirical grounds for suggesting that CMM is a better measure for understanding the biomedical effects of co-occurring chronic diseases. This hypothesis was partially supported by the finding that CMM was statistically associated with mortality [27]. In addition, previous studies showed that CMM was a risk factor for the need for long-term care. On the other hand, those studies evaluated the impact of CMM on both acute and chronic disease care [28]. Considering that CMM better reflects chronic disease burdens, we think it is reasonable to incorporate CMM in determining eligibility for LTC services [29].

There are several limitations to this study. Firstly, disease information on the participants was obtained by self-report, which may be subject to information bias (i.e., respondents not being aware of their disease status, or misreporting their diagnoses), although some papers recommend that data collection be carried out via self-report in multimorbidity studies, even for older adults [30]. Secondly, information on diseases was assessed only at baseline and not updated over the six-year follow-up. Third, we used 17 diseases surveyed through JAGES for the analysis. Given that this does not cover all diseases, our findings may underestimate the prevalence of MM/CMM and their influence on incident LTC needs. Lastly, given its observational design, this study does not prove causation between multimorbidity and the LTC insurance certification.

However, we believe that this study has strengths over other studies on several points. The propensity score method was used for the analysis to adjust as many covariates as possible to assess the CMM effect. Moreover, despite a nation-wide cohort study involving a large number of participants, it achieved a high follow-up rate (96.2%) after the six-year follow-up.

## 5. Conclusions

Both MM and CMM were associated with a higher risk of LTC insurance certification in Japan. These data will hopefully provide useful information in the assessment of the need for LTC.

## Figures and Tables

**Figure 1 ijerph-18-10523-f001:**
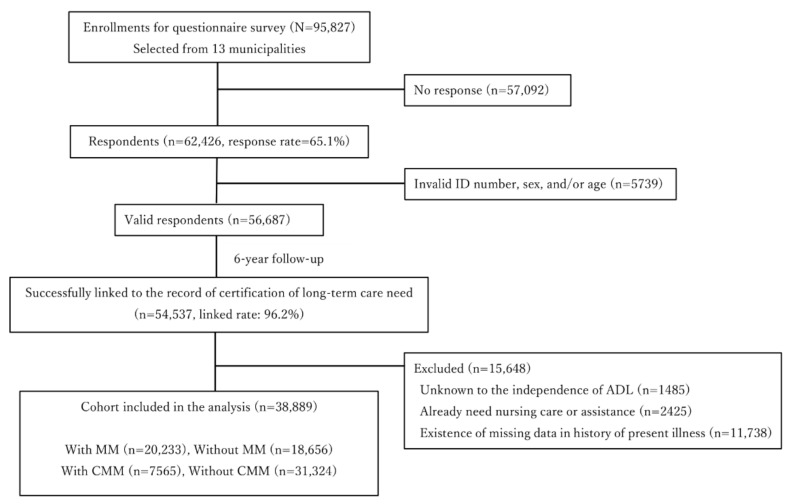
Study Population. Complex multimorbidity (CMM), multimorbidity (MM,).

**Figure 2 ijerph-18-10523-f002:**
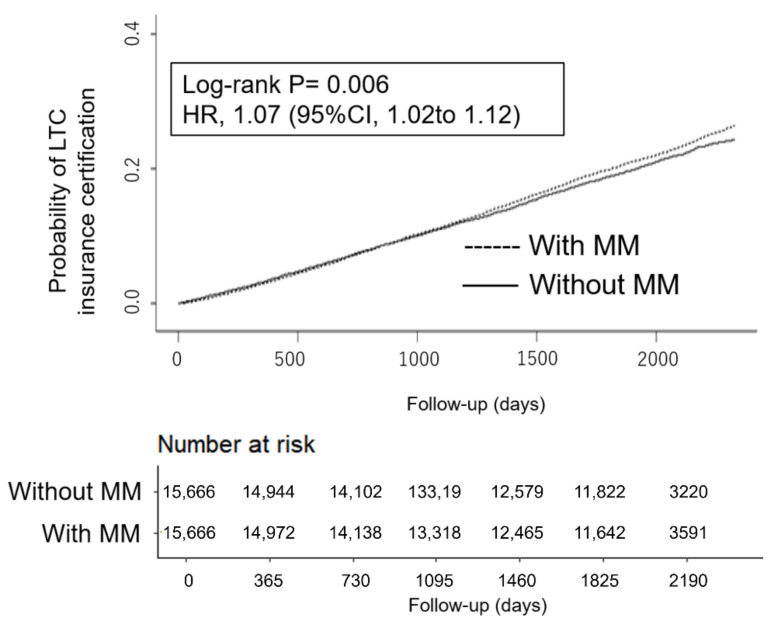
Cumulative incidence curve for overall care needs between patients with the MM group and patients without the MM group. Confidence interval (CI), hazard ratio; long-term care (LTC); multimorbidity (MM).

**Figure 3 ijerph-18-10523-f003:**
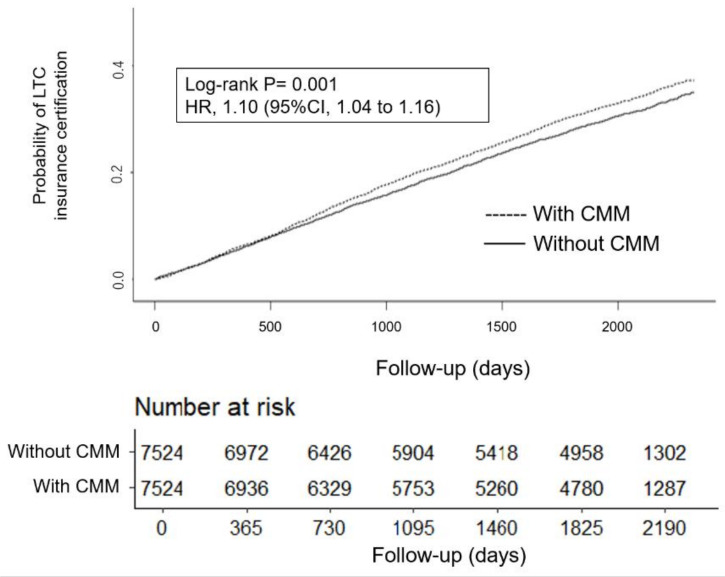
Cumulative incidence curve for overall care needs between patients with the CMM group and patients without the CMM group. Confidence interval (CI), complex multimorbidity (CMM); hazard ratio (HR), long-term care (LTC).

**Table 1 ijerph-18-10523-t001:** Definition of CMM ^a^.

Category	Disease
Circulation disorder	Heart disease (including arrhythmia)
	Stroke
	High blood pressure
Endocrine–metabolic disorder (General system)	Diabetes (including mild-type)
	Obesity
	Dyslipidemia
Eye disorder	Impaired vision
Gastrointestinal disorder	Gastrointestinal disease
	Liver disease
Hearing disorder	Impaired hearing
Mental and behavioral disorder	Mental disease
	Sleep problem
Musculoskeletal and connective disorder	Osteoporosis
	Joint disease/Neuralgia
	Injury/Fracture
Neoplasm	Cancer
Respiratory disorder	Respiratory disease

^a^ Complex multimorbidity (CMM).

**Table 2 ijerph-18-10523-t002:** Standardized mean differences with or without MM ^a^, before and after Propensity Score Matching.

Characteristic	SMD ^b^ in Multiply Imputed Data	SMD in Matching Data
Age	0.24	0.002
Sex	0.099	0.001
Previous health check-up	0.01	0.015
The number of natural teeth	0.11	0.019
Consumption of meat and fish	0.009	0.017
Consumption of fruits and vegetables	0.003	0.006
Formal educational year	0.093	0.045
Marital status	0.072	0.015
Someone living with you	0.06	0.033
Residence type	0.025	0.055
Architectural type of home	0.005	0.086
Worries about unexpected expenses	0.123	0.004
Receiving pension	0.023	0.022
Current working status	0.147	0.002
Persons to have meal with	0.089	0.02
Alcohol	0.107	0.015
Smoking	0.079	0.014
Falling over	0.223	0.004
Worriees about falls	0.266	0.001
Going upstairs without support	0.265	0.009
Getting up out of chairs without support	0.251	0.02
Average time walking	0.16	<0.001
Frequency of going out	0.151	0.015
Decrease in the frequency of going out	0.243	0.001
Engagement in leisure activities	0.105	0.016
Trust in neighbors	0.079	0.027
Support from neighbors	0.074	0.015
Attachment to residence	0.053	0.036
Contribution to residence	0.095	0.009
Uneasiness about safety in residence	0.073	0.011
Participation in local events	0.085	0.009
Interactions with neighborhood	0.02	0.031
Near to residence		
Locations with graffiti or garbage	0.009	0.02
Parks or foot paths	0.059	0.045
Locations difficult for walking	0.076	0.012
Risky roads or crossroads for traffic accidents	0.044	0.005
Fascinating views or buildings	0.04	<0.001
Shops selling fresh fruits and vegetables	0.074	0.023
Dangerous places walking alone at night	0.013	0.019
Comfortable houses or facilities	0.066	0.024
Someone listening to your concerns	0.019	0.01
Someone looking after you in the case of illness	0.049	0.023
Attendance		
Sports group or club	0.063	0.008
Leisure activity group	0.06	0.006

^a^ multimorbidity (MM), ^b^ standardized mean difference (SMD).

**Table 3 ijerph-18-10523-t003:** Standardized mean differences with or without CMM ^a^, before and after Propensity Score Matching.

Characteristic	SMD ^b^ in Multiply Imputed Data	SMD in Matching Data
Age	0.327	0.025
Sex	0.139	0.004
Previous health check-up	0.02	0.005
The number of natural teeth	0.16	0.005
Consumption of meat and fish	0.017	0.016
Consumption of fruits and vegetables	0.035	0.012
Formal educational year	0.151	0.004
Marital status	0.118	0.002
Someone living with you	0.1	0.011
Residence type	0.058	0.008
Architectural type of home	0.02	0.006
Worries about unexpected expenses	0.21	0.012
Receiving pension	0.022	0.006
Current working status	0.225	0.004
Persons to have meal with	0.17	0.014
Alcohol	0.145	0.013
Smoking	0.098	0.016
Falling over	0.307	0.013
Worries about falls	0.396	0.005
Going upstairs without support	0.348	0.005
Getting up out of chairs without support	0.343	0.01
Average time walking	0.203	0.001
Frequency of going out	0.207	0.003
Decrease in the frequency of going out	0.352	0.006
Engagement in leisure activities	0.145	0.008
Trust in neighbors	0.135	0.009
Support from neighbors	0.109	0.002
Attachment to residence	0.086	0.002
Contribution to residence	0.129	0.007
Uneasiness about safety in residence	0.105	0.01
Participation in local events	0.114	0.008
Interactions with neighborhood	0.049	0.007
Near to residence		
Locations with graffiti or garbage	0.019	0.014
Parks or foot paths	0.097	<0.001
Locations difficult for walking	0.132	0.007
Risky roads or crossroads for traffic accidents	0.061	0.002
Fascinating views or buildings	0.074	0.004
Shops selling fresh fruits and vegetables	0.091	0.001
Dangerous places walking alone at night	0.016	<0.001
Comfortable houses or facilities	0.107	0.011
Someone listening to your concerns	0.075	0.007
Someone looking after you in the case of illness	0.094	0.026
Attendance		
Sports group or club	0.117	0.031
Leisure activity group	0.088	0.007

^a^ complex multimorbidity (CMM); ^b^ standardized mean difference (SMD).

## Data Availability

Data are from the JAGES study. All inquiries are to be addressed to the data management committee via e-mail: dataadmin.ml@jages.net. All JAGES datasets have ethical and legal restrictions on public deposition due to the inclusion of sensitive information from human participants.

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
