# Peer review of "Complex Multimorbidity and Incidence of Long-Term Care Needs in Japan: A Prospective Cohort Study"

_ijerph, 2021, doi:10.3390/ijerph181910523_

Round 1

Reviewer 1 Report

My main and only concern with the manuscript is the attention given to its limitations in contrast with the actual findings. For instance, the authors dedicated 5 paragraphs in the discussion to point out and discuss major limitations, which are all legit. Furthermore, if the authors themselves think that there are more limitations than benefits, maybe the actual value of the manuscript is questionable. Maybe the authors can expand on what their research contributes to the field more than focusing on the negative aspects of it.

Author Response

We appreciate your review and thoughtful comments. We have revised the manuscript on the basis of your comments.

Reviewer’s comment #1

My main and only concern with the manuscript is the attention given to its limitations in contrast with the actual findings. For instance, the authors dedicated 5 paragraphs in the discussion to point out and discuss major limitations, which are all legit. Furthermore, if the authors themselves think that there are more limitations than benefits, maybe the actual value of the manuscript is questionable. Maybe the authors can expand on what their research contributes to the field more than focusing on the negative aspects of it.

Response #1

Thank you for your comment. As below, we have reduced the description of limitations and instead increased the description of the significance of this study.

line 257-261

The results of our study suggest the importance of focusing on morbidity across several body systems (i.e., complex multimorbidity) in order to prevent disability. Furthermore, when combined with our previous findings, [26] our results suggest that older individuals with MM/CMM are at increased risk of long-term care needs as well as higher mortality.

line 273-282

There are several limitations to this study. Firstly, disease information of the participants was obtained from self-report, which may be subject to information bias, (i.e., respondents not being aware of their disease status, or misreporting their diagnoses), although some papers recommend the data collection via self-report in multimorbidity studies, even for older adults. [30] Secondly, information on diseases was assessed only at baseline and not updated over the six-year follow-up. Third, we used 17 diseases surveyed through JAGES for analysis. Given this does not cover all diseases, our findings may underestimate the MM/CMM prevalence and their influence on incident LTC needs. Lastly, given its observational design, this study does not prove causation between multimorbidity and the LTC insurance certification.

Reviewer 2 Report

Thank you for the opportunity to review the manuscript entitled "Complex multimorbidity and incidence of long-term care needs in Japan: a prospective cohort study" -- I have included suggested revisions below and I would be happy to review a resubmission. 

Abstract
-Please change "an improved" to "a more nuanced"
-Please change "over 65" to "over 65 years"

Introduction
-Please change "multi-morbidity" to "multimorbidity" throughout

Methods
-Please change "over 65" to "over 65 years"
-Please change "populations" to "individuals" 
-Please change "comorbidity and downstream" to "multimorbidity and downstream"

Results
-Please change "more likely to get married" to "more likely to be married"

Discussion
-Please change "gender" to "sex" throughout
-Please change "in the older adults" to "in older adults"
-Please change "biomedical effects of comorbidity" to "biomedical effects of co-occurring chronic diseases"

Author Response

We appreciate your thoughtful review. We revised the manuscript based on your comments.

Reviewer’s comment #1

Abstract

-Please change "an improved" to "a more nuanced"

-Please change "over 65" to "over 65 years"

Introduction

-Please change "multi-morbidity" to "multimorbidity" throughout

Methods

-Please change "over 65" to "over 65 years"

-Please change "populations" to "individuals"

-Please change "comorbidity and downstream" to "multimorbidity and downstream"

Results

-Please change "more likely to get married" to "more likely to be married"

Discussion

-Please change "gender" to "sex" throughout

-Please change "in the older adults" to "in older adults"

-Please change "biomedical effects of comorbidity" to "biomedical effects of co-occurring chronic diseases"

Response #1

Thank you for your comment. We fixed every point you requested.

Reviewer 3 Report

Firstly, the authors are interested to explore the association of multimorbidity with the incidence of long term care needs for older adults in Japan with two conventional definition of multimorbidity and complex multimorbidity. Overall, I thought the study was well designed with backing of JAGES study and delicate statistical analysis using propensity score matching and IPTW. 

Just curious, how did the study team come up with the definition of CMM? I looked at Table 1 and puzzled with circulation disorder and endocrine-metabolic disorder being parked in different categories. I believe the chronic conditions in both categories are concordant and should be merged as cardiometabolic disorders. According to a systematic review published on patterns of associative multimorbidity in Asia {Rajoo SS, Wee ZJ, Lee PSS, Wong FY, Lee ES. A Systematic Review of the Patterns of Associative Multimorbidity in Asia. Biomed Res Int. 2021 Jul 3;2021:6621785. doi: 10.1155/2021/6621785}, there were 5 main categories of patterns which are quite similar to your study for your reference.

I noticed that impaired vision and impaired hearing are categorised as eye disorder and hearing disorder. I would have thought it may make more sense to categorise them as physical disability instead. Did you include cataract under eye disorder?

For mental disease, joint or respiratory diseases, would it be any mental or joint or respiratory diseases the participant self-report? 

Just thought the presentation of the results (in the supplementary section) are a little messy as there are many variables being included.

Discussion section was a little short. Would love to see more on the impact of this research on the elderly population and how these findings can be translated to other elderly population outside Japan who may face similar issues.

Minor point: there was inconsistencies in spelling multimorbidity, with and without hyphen eg. line 33, 56, 132. There was an error with reference 11 at line 57.

Author Response

We appreciate your review and thoughtful comments. We have revised the manuscript on the basis of your comments.

Reviewer’s comment #1

Just curious, how did the study team come up with the definition of CMM? I looked at Table 1 and puzzled with circulation disorder and endocrine-metabolic disorder being parked in different categories. I believe the chronic conditions in both categories are concordant and should be merged as cardiometabolic disorders. According to a systematic review published on patterns of associative multimorbidity in Asia {Rajoo SS, Wee ZJ, Lee PSS, Wong FY, Lee ES. A Systematic Review of the Patterns of Associative Multimorbidity in Asia. Biomed Res Int. 2021 Jul 3;2021:6621785. doi: 10.1155/2021/6621785}, there were 5 main categories of patterns which are quite similar to your study for your reference.

Response #1

Thank you for your comment. In this study, we followed ICD-10 to determine which body system is impaired by each disease. This is the approach used in the original paper in which the concept of CMM was proposed, (1) and has already been adopted in several CMM studies. (2)(3)

<Reference>

  1. Harrison, Christopher, et al. "Examining different measures of multimorbidity, using a large prospective cross-sectional study in Australian general practice." BMJ open4.7 (2014): e004694.

  1. Storeng, Siri H., et al. "Associations between complex multimorbidity, activities of daily living and mortality among older Norwegians. A prospective cohort study: The HUNT Study, Norway." BMC geriatrics20.1 (2020): 1-8.
  2. Vinjerui, Kristin Hestmann, et al. "Socioeconomic inequalities in the prevalence of complex multimorbidity in a Norwegian population: findings from the cross-sectional HUNT Study." BMJ open10.6 (2020): e036851.

Reviewer’s comment #2

I noticed that impaired vision and impaired hearing are categorised as eye disorder and hearing disorder. I would have thought it may make more sense to categorise them as physical disability instead. Did you include cataract under eye disorder?

Response #2

Thank you for your comment. As mentioned in #2, in this study, the disorders were classified according to ICD-10. Therefore, we are concerned that lumping eye disorder and hearing disorder together and regarding them as physical disabilities may be a bit crude. Yes, cataract is classified as an eye disorder.

Reviewer’s comment #3

For mental disease, joint or respiratory diseases, would it be any mental or joint or respiratory diseases the participant self-report?

Response #3

Thank you for your comment. Yes, the Japan Gerontological Evaluation Study (JAGES), the source of the data we analyzed, collected the data in that way.

Reviewer’s comment #4

Just thought the presentation of the results (in the supplementary section) are a little messy as there are many variables being included.

Response #4

Thank you for your comment. We have changed the fonts to improve readability.

Reviewer’s comment #4

Discussion section was a little short. Would love to see more on the impact of this research on the elderly population and how these findings can be translated to other elderly population outside Japan who may face similar issues.

Response #4

Thank you for your comment. We have added sentences about the significance of this study results as follows.

line 257-261

The results of our study suggest the importance of focusing on morbidity across several body systems (i.e., complex multimorbidity) in order to prevent disability. Furthermore, when combined with our previous findings, [26] our results suggest that older individuals with MM/CMM are at increased risk of long-term care needs as well as higher mortality.

Reviewer’s comment #4

Minor point: there was inconsistencies in spelling multimorbidity, with and without hyphen eg. line 33, 56, 132. There was an error with reference 11 at line 57.

Response #4

Thank you for your comment. We fixed all of them.